# A Phase 2a, Randomized, Double-Blind, Placebo-Controlled Study to Assess the Efficacy and Safety of ALS-L1023 in Non-Alcoholic Fatty Liver Disease

**DOI:** 10.3390/ph16040623

**Published:** 2023-04-20

**Authors:** Gi-Ae Kim, Hyun Chin Cho, Soung Won Jeong, Bo-Kyeong Kang, Mimi Kim, Seungwon Jung, Jungwook Hwang, Eileen L. Yoon, Dae Won Jun

**Affiliations:** 1Department of Internal Medicine, Kyung Hee University College of Medicine, Kyung Hee University Hospital, Seoul 02447, Republic of Korea; antiankle@naver.com; 2Department of Internal Medicine, Gyeongsang National University School of Medicine and Gyeongsang National University Hospital, Jinju 52727, Republic of Korea; 3Department of Internal Medicine, Soonchunhyang University College of Medicine, Soonchunhyang University Seoul Hospital, Seoul 04401, Republic of Korea; 4Department of Radiology, Hanyang University College of Medicine, Seoul 04763, Republic of Korea; 5Graduate School for Biomedical Science & Engineering, Hanyang University, Seoul 04763, Republic of Korea; 6Hanyang Institute of Bioscience and Biotechnology, Hanyang University, Seoul 04763, Republic of Korea; 7Department of Internal Medicine, Hanyang University College of Medicine, Seoul 04763, Republic of Korea

**Keywords:** ALS-L1023, non-alcoholic fatty liver disease, non-alcoholic steatohepatitis

## Abstract

Preclinical data have shown that the herbal extract, ALS-L1023, from *Melissa officinalis* reduces visceral fat and hepatic steatosis. We aimed to assess the safety and efficacy of ALS-L1023 as the treatment of non-alcoholic fatty liver disease (NAFLD). We conducted a 24-week randomized, double-blind, placebo-controlled 2a study in patients with NAFLD (MRI-proton density fat fraction [MRI-PDFF] ≥ 8% and liver fibrosis ≥ 2.5 kPa on MR elastography [MRE]) in Korea. Patients were randomly assigned to 1800 mg ALS-L1023 (*n* = 19), 1200 mg ALS-L1023 (*n* = 21), or placebo (*n* = 17) groups. Efficacy endpoints included changes in liver fat on MRI-PDFF, liver stiffness on MRE, and liver enzymes. For the full analysis set, a relative hepatic fat reduction from baseline was significant in the 1800 mg ALS-L1023 group (−15.0%, *p* = 0.03). There was a significant reduction in liver stiffness from baseline in the 1200 mg ALS-L1023 group (−10.7%, *p* = 0.03). Serum alanine aminotransferase decreased by −12.4% in the 1800 mg ALS-L1023 group, −29.8% in the 1200 mg ALS-L1023 group, and −4.9% in the placebo group. ALS-L1023 was well tolerated and there were no differences in the incidence of adverse events among the study groups. ALS-L1023 could reduce hepatic fat content in patients with NAFLD.

## 1. Introduction

Non-alcoholic fatty liver disease (NAFLD) is one of the most common chronic liver diseases worldwide as its prevalence in the general population is 25% [1,2]. The global burden of NAFLD is on the rise with the increasing prevalence of obesity [1,3,4]. Non-alcoholic steatohepatitis (NASH) is a progressive form of NAFLD that occurs in close association with obesity, type 2 diabetes, and metabolic syndrome [3,5,6]. No approved pharmacological options are currently available for patients with NASH; therefore, the demand for developing new therapeutic agents is still high.

It has been reported that angiogenesis is a key mechanism in the progression of NAFLD, and angiogenesis drives inflammation and liver fibrosis in NAFLD [7,8]. An herbal extract, ALS-L1023, from *Melissa officinalis* is suggested to have an anti-angiogenic effect and to be capable of reducing adipose tissue mass in high-fat diet-induced obese mice [9]. Several studies have shown that ALS-L1023 can regulate not only obesity but also hepatic steatosis and inflammation in obese NAFLD mice [10,11]. Studies have found that it has antioxidant activity and protects cells against oxidative stress-induced apoptosis [11]. Preclinical data also suggested that ALS-L1023 can have the potential of improving visceral obesity and insulin resistance which may contribute to the reduction of inflammation in the liver [12,13,14]. Thus, studies that investigate the anti-angiogenic potentials of ALS-L1023 as a pharmacological treatment option for patients with NAFLD are required.

In this randomized, placebo-controlled, phase 2a clinical trial, we aimed to assess the efficacy and safety of ALS-L1023 as the treatment of NAFLD and investigate the clinically appropriate administration dose of ALS-L1023.

## 2. Results

### 2.1. Baseline Characteristics

A total of 138 patients were screened at 4 sites, and 60 patients who satisfied the eligibility criteria were enrolled and randomly assigned to the 1800 mg ALS-L1023 (*n* = 19), 1200 mg ALS-L1023 (*n* = 21), or placebo (*n* = 20) groups (Figure 1 and Appendix A). All patients who received at least one dose of the study drug after randomization were included in the safety analysis (safety set, *n* = 60; Table 1). Patients with the full analysis data after administration comprised 95% of the safety set (full analysis set [FAS], *n* = 57). Of the FAS, a total of 55 patients completed the administration and had MRI-proton density fat fraction (MRI-PDFF) measurements at both baseline and week 24. Patients who dropped out (*n* = 1), had poor compliance of less than 80% (*n* = 1), and had protocol deviation that included taking medications that should not be taken together (*n* = 5) were further excluded (per protocol set [PPS], *n* = 50; Table 1).

### 2.2. Changes in Hepatic Steatosis on MRI-PDFF

Figure 2A and Table 2 show the changes in hepatic steatosis from baseline to week 24 on MRI-PDFF. In the FAS, greater median relative fat reductions were observed in the 1800 mg ALS-L1023 group (−15.0% vs. 0%, *p* = 0.95) and in the 1200 mg ALS-L1023 group (−6.9% vs. 0%, *p* = 0.65), compared to those of the placebo group. When the hepatic steatosis of week 24 was compared to that of baseline, statistically significant differences were observed in the 1800 mg ALS-L1023 group (−15.0%; *p* = 0.03). Greater median absolute fat reductions at week 24 were observed in the 1800 mg ALS-L1023 group (−1.6% vs. 0%, *p* = 0.48) and the 1200 mg ALS-L1023 group (−1.5% vs. 0%, *p* = 0.85) compared to the placebo group (Figure 2B). The number of patients with ≥20% relative fat reduction on MRI-PDFF at week 24 were seven (38.9%) in the 1800 mg ALS-L1023 group, eight (40.0%) in the 1200 mg ALS-L1023 group, and five (29.4%) in the placebo group (Table 2). The number of patients with ≥15% relative fat reduction was nine (50.0%) in the 1800 mg ALS-L1023 group, nine (45.0%) in the 1200 mg ALS-L1023 group, and five (29.4%) in the placebo group.

For the PPS, compared with the placebo group, greater median relative fat reductions were observed in the 1800 mg ALS-L1023 group (−15.0% vs. 0.3%, *p* = 0.49) and the 1200 mg ALS-L1023 group (−9.5% vs. 0.3%, *p* = 0.99; Appendix A). Greater median absolute fat reductions at week 24 were observed in the 1800 mg ALS-L1023 group (−1.6% vs. 0.1%) and the 1200 mg ALS-L1023 group (−1.9% vs. 0.1%) than in the placebo group.

### 2.3. Changes in Liver Stiffness on MRE

Changes in liver stiffness on MR elastography (MRE) from baseline to week 24 are shown in Table 3. For the FAS, relative reductions in liver stiffness observed in the study groups were −6.2% in the 1800 mg ALS-L1023 group, −10.7% in the 1200 mg ALS-L1023 group, and −7.6% in the placebo group. A significant relative reduction (−10.7%, *p* = 0.03) and absolute reduction (−0.4 kpa, *p* = 0.02) in liver stiffness were observed in the 1200 mg ALS-L1023 group. For the PPS, there were no significant differences in the relative and absolute reduction of liver stiffness among the three groups.

### 2.4. Changes in Liver Biochemistry

In the FAS, changes in serum alanine aminotransferase (ALT) from baseline to week 24 were found in the 1800 mg group (−12.4% vs. −4.9%; *p* = 0.11) and in the 1200 mg group (−29.8% vs. −4.9%; *p* = 0.10), compared to the placebo group (Table 4, Figure 3A). In the PPS, significant reductions in serum ALT from baseline to week 24 were found in the 1800 mg group (−13.5% vs. 0.7%; *p* = 0.04) and in the 1200 mg group (−36.0% vs. 0.7%; *p* = 0.03), compared to the placebo group (Appendix A). Reductions by week in the FAS and PPS are shown in Appendix A. In the FAS, changes in serum aspartate aminotransferase (AST) from baseline to week 24 were found in the 1200 mg group (−19.6% vs. 0%; *p* = 0.12), compared to the placebo group (Table 4, Figure 3B). In the PPS, significant reductions in serum AST from baseline to week 24 were found in the 1200 mg group (−21.0% vs. 0%; *p* = 0.15), compared to the placebo group (Appendix A). In the FAS, changes in serum total cholesterol from baseline to week 24 were found in the 1200 mg group (−5.5% vs. 5.4%, *p* = 0.03; Table 4, Figure 3C), compared to the placebo group. For the PPS, the reduction in total cholesterol from baseline to week 24 was found in the 1200 mg group (−5.8% vs. 5.4%, *p* < 0.01; Appendix A). The changes in other liver biochemical parameters are summarized in Table 4 and Appendix A.

### 2.5. Safety Results

The most commonly reported adverse events were gastrointestinal conditions including dyspepsia, abdominal distension, and abdominal pain (Table 5). Most adverse events were mild and not severe enough to necessitate the discontinuation of treatment. No significant differences were observed between the groups. One patient in the placebo group had a serious adverse event due to acute myocardial infarction; however, treatment with the study drug was not discontinued since a causal relationship between the study drug and the adverse event was not found.

### 2.6. Selected miRNAs and the Putative Transcripts

Several studies have identified enriched miRNAs in patients with NAFLD. To determine the levels of circulating miRNAs in the patient serum of ALS-L1023 treated patients, we performed miRNA microarray analysis and quantified the levels of miRNAs. We then selected 23 miRNAs that were significantly altered by more than 1.5-fold in the ALS-L1023 treatment groups compared with the placebo group (Figure 4A and Appendix A). Interestingly, all 23 miRNAs were downregulated by ALS-L1023 treatment. The numbers of significant miRNAs from each study group are presented in Figure 4B and the number of putative miRNAs was six. We verified 6 functional miRNAs using the TargetScan algorithm (https://www.targetscan.org/, accessed on 1 April 2023) and identified putative 499 miRNAs target transcripts, assuming that the levels of putative targets were upregulated by the downregulation of miRNAs (Appendix A). To assess the cellular signaling involved in the target transcripts, we performed the KEGG analysis, which suggested 19 significantly enriched pathways (Figure 4C and Appendix A). It is well known that most of the nine enriched pathways shown in Figure 4C are closely related to the pathogenesis of NAFLD. In particular, insulin resistance, Wnt signaling, mTOR pathway, TGF-beta signaling, focal adhesion, MAPK, and PI3K-Akt signaling pathways are strongly associated with steatosis and hepatic fibrosis.

## 3. Discussion

This clinical trial is the first to apply an angiogenesis inhibitor as a NAFLD treatment. A phase 2a, multicenter, placebo-controlled, randomized clinical trial was conducted to evaluate the safety and exploratory efficacy of 1200 mg and 1800 mg ALS-L1023 in patients with NAFLD to determine a clinically appropriate dose. The 24-week treatment with ALS-L1023 resulted in a dose-dependent decrease in hepatic steatosis on MRI-PDFF. The 1200 mg ALS-L1023 group showed significant reductions in liver stiffness, total cholesterol, and LDL-cholesterol levels within the group after treatment. Serum ALT decreased in both the 1800 mg and 1200 mg ALS-L1023 groups. ALS-L1023 was well tolerated and there were no significant adverse effects related to the administration of the study drug compared to the placebo group.

ALS-L1023 is a fractional extract from *Melissa officinalis* and has anti-angiogenic and antioxidant activities, which prevent oxidative stress-induced apoptosis [10,11,13,14]. It has been reported that angiogenesis is closely associated with the progression of liver fibrosis and HCC, both in NAFLD as well as other chronic liver diseases [8,9]. A previous animal study showed that ALS-L1023 can regulate obesity and reduce hepatic steatosis and inflammation in obese NAFLD mice [11]. It has also been reported that ALS-L1023 contributes to improving visceral obesity and insulin resistance [14]. Given these preclinical data and complementary mechanisms of ALS-L1023, it would be reasonable to explore the potential of ALS-L1023 in treating patients with NAFLD, considering the heterogeneous character of NAFLD. Indeed, in this phase 2a clinical trial, ALS-L1023 administration was found to be beneficial in patients with NAFLD as it reduced liver fat and fibrosis.

Our data showed that patients treated with 1800 mg ALS-L1023 had significantly reduced hepatic steatosis on MRI-PDFF after 24 weeks of administration. Several recent studies have shown that significant fat reduction is associated with histologic response and NASH resolution [15,16,17]. Although high-dose ALS-L1203 did not reduce the hepatic fibrosis burden after 24 weeks of treatment, low-dose ALS-L1203 showed a significant reduction in liver fibrosis. The lack of treatment response seemed to originate from the small number of patients in this study. With a larger number of study patients and long-term administration, further reduction in hepatic steatosis and fibrosis can be expected. 

Although the ALT reduction in the FAS was not significant at week 24, the reduction was significant in the PPS for both 1200 mg and 1800 mg ALS-L1023 groups compared to the placebo group. In the FAS, the ALT levels decreased significantly at week 8 and 16 in the 1200 mg ALS-L1023 group, and week 16 in the 1800 mg ALS-L1023 group compared to the placebo group. Other markers of liver inflammation and injury, including AST and GGT, also decreased after the administration of ALS-L1023. However, biomarkers such as hepatic fibrogenesis (N-terminal propeptide of type 3 procollagen [PRO-C3]) and ballooning (cytokeratin-18 [CK-18]) showed no improvement after the administration of ALS-L1023. The miRNA analysis of the study patients showed that 23 miRNAs were downregulated by ALS-L1023 and GO analysis using the putative miRNA targets suggested that ALS-L1023 may have effects on the pathogenesis of NAFLD through the enriched signaling. ALS-L1023 treatment could have regulated the abundance of miRNAs in the serum, thereby resulting in changes in cellular signaling.

ALS-L1023 was well tolerated in patients with NAFLD and had mild-to-moderate adverse effects. No treatment discontinuation due to adverse events occurred in patients treated with ALS-L1023; however, one patient in the placebo group had to discontinue treatment because of grade 3–4 laboratory abnormalities. The most frequent adverse events reported were gastrointestinal conditions such as dyspepsia, abdominal distension, and abdominal pain. These conditions were mild and self-limiting and did not result in study withdrawal because the patients spontaneously recovered. ALS-L1023 can serve as a safe pharmacological treatment option for patients with NAFLD, without causing significant adverse events.

This study has several limitations. First, the small number of patients included in each study group can challenge the generalization of the findings. Therefore, the efficacy and safety observed in this study should be interpreted with caution. However, the number of patients included in this study satisfies the exploratory nature of the study. Subgroup analysis by gender could not be performed since the number of patients was rather small. Second, the diagnosis of NAFLD was made based on non-invasive assessments of MRI-PDFF and MRE rather than histological assessment. Given the reported association between non-invasive parameters and histologic response, using MRI-PDFF or MRE can be a good alternative for assessing liver fat and fibrosis [16,17,18,19], especially under resource-limited circumstances and considering the invasiveness of liver biopsy. Finally, the patients who participated in this study were free of specific lifestyle guidance, such as controlling their diet or exercising. Lifestyle modifications were not monitored during the study period. Thus, this study did not take into account the possible influence of lifestyle factors on treatment efficacy.

## 4. Materials and Methods

### 4.1. Study Design

This multicentre, randomized, double-blind, placebo-controlled, phase 2a study was conducted at four academic centers in the Republic of Korea. The trial protocol was approved by the institutional review board at each center, and this trial has been registered on cris.nih.go.kr (registration number, KCT0005256, http://cris.nih.go.kr, accessed on 1 April 2023).

### 4.2. Inclusion and Exclusion Criteria

Patients who met the following criteria were included in this study: male or female patients aged 19 years or older but younger than 75 years; patients with a clinical diagnosis of NAFLD, defined as steatosis ≥ 8% on MRI-PDFF and liver stiffness ≥ 2.5 kPa on MRE within 3 months of screening. Key exclusion criteria were excessive alcohol consumption (>210 g per week for male or >140 g per week for female patients), history of viral hepatitis, decompensated liver disease, hepatocellular carcinoma, other causes of liver disease, uncontrolled hypertension, unstable cardiovascular disease, malignancy, ALT ≥ 5x the upper limit of normal, and HbA1c > 9.0%. Patients with a history of glucagon-like peptide (GLP)-1 receptor agonist treatment within 8 weeks prior to screening were also excluded. Patients who received a high dose of vitamin E (>400 IU/day) or thiazolidinediones were not excluded when they were on a stable dose without any change in dose for at least 180 days prior to the screening. All inclusion and exclusion criteria are provided in the Appendix A. All participants provided written informed consent prior to enrolment.

### 4.3. Procedures

The patients were randomly assigned to either the 1800 mg ALS-L1023, 1200 mg ALS-L1023, or placebo groups in a 1:1:1 ratio. SAS V9.4 or higher was used for randomization and treatment assignment. The trial consisted of a 4-week screening and 24-week treatment periods (Appendix A). Patients were instructed to administer three tablets of ALS-L1023 twice daily with or without food. The screening assessments included medical history, physical examination, standard laboratory test results, and imaging assessments. Imaging assessments of steatosis on MRI-PDFF and liver stiffness on MRE were conducted at screening and at week 24. MRI-PDFF and MRE images were analyzed using a central reader. Serum samples for clinical laboratory parameters and serum biomarkers were collected at baseline and every 8 weeks until the end of this study, and analyzed at the central laboratory (SCL Healthcare, Seoul, Republic of Korea).

### 4.4. Outcomes

Efficacy endpoints included changes in liver fat on MRI-PDFF, changes in liver stiffness on MRE at week 24 compared to the screening, changes in visceral fat at week 24 compared to the screening, changes in ALT at week 24 compared to baseline, and changes in AST at week 24 compared to baseline. Exploratory endpoints included the following: changes in ALT at weeks 8 and 16; changes in AST at weeks 8 and 16; changes in triglycerides at weeks 8 and 16; changes in total cholesterol at weeks 8, 16, and 24; and changes in Pro-C3, CK-18, HOMA-IR, Leptin, Ghrelin, and adiponectin at week 24. The safety assessments included clinical laboratory tests, vital signs, electrocardiograms, physical examinations, and adverse events. Clinical and laboratory adverse events were coded using the Medical Dictionary for Regulatory Activities version 23.0.

### 4.5. Assessment of MRI-PDFF and MRE

Image assessment was performed by two experienced central readers who were unaware of clinical, histological data and treatment assignments. All sites underwent a quality assessment process prior to study initiation based on a review of hardware, software, phantom, and volunteer scans. All research images obtained from the trial were approved by the central readers. Liver stiffness was estimated using a gradient-echo motion detection sequence. A 60 Hz oscillation was transmitted to the liver by a driver centered on the right midclavicular line at the xiphoid level and held in place by an elastic band. This image was converted to an elastogram using an inversion algorithm that represents the estimated abdominal tissue stiffness in pixels. Three or four separate images were obtained. The central readers manually drew a region of interest (ROI) on the elastogram results of each slice to include as much liver tissue as possible, where a consistent shear wave is visible. The mean stiffness value of the liver was calculated and used [20]. For fat measurement, three-dimensional volumetric chemical shift encoded MRI image data were acquired in the axial orientation. Quantitative parametric PDFF maps were reconstructed using the online software of each vendor, accounting for multi-peak fat spectral modeling and correcting for T2* decay. Data analysts placed two circular ROIs with a radius of 1 cm in the nine anatomical liver segments. The average PDFF value for the whole liver was calculated as the average of the evaluable PDFF values from nine segments.

### 4.6. miRNA Microarray

The integrity of the extracted total RNA was checked using an Agilent Bioanalyzer^®^ (Agilent Technologies, Santa Clara, CA, USA). A fluorescent sample was attached to the RNA and hybridization was performed on human miRNA microarrays (Agilent Technologies) containing 13,737 probes corresponding to 799 mature microRNAs and 22 control probes. For the cDNA microarray analysis, 50 ng of RNA was hybridized to Agilent 44 K oligonucleotide microarrays (Agilent Technologies). After correcting with control probes for each chip, those with a *p*-value ≤ 0.05 were selected.

### 4.7. Full Analysis Set and per Protocol Set

The safety set included all patients who received at least one dose of the study drug after randomization. The demographic data and baseline characteristics of the patients were analyzed using the safety set. All efficacy analyses were performed on the FAS and PPS. The FAS was defined as patients with full analysis data after administration, and the PPS comprised all patients who completed this study without major protocol deviations and had an average drug compliance of 80% or higher. When assessing the efficacy points, missing data were imputed using the last-observation-carried-forward approach (LOCF) in the FAS analysis. However, LOCF was not applied if there were no MRI-PDFF and MRE data at week 24.

### 4.8. Statistical Analysis

Since this was an exploratory study, a formal power calculation was not conducted when determining the sample size. The sample size assessment was based on other proof-of-concept studies that assessed directionality across multiple outcomes, rather than statistical significance. Comparative analyses between the groups were conducted. Continuous variables were analyzed using an analysis of variance (ANOVA) or the Kruskal–Wallis test. Categorical variables were analyzed using a Fisher’s exact test or the chi-squared test. Efficacy endpoints between the groups were compared by a Two-sample *t*-test or Wilcoxon rank-sum test. All analyses were based on observed data and were performed using SAS version 9.4 (SAS; Cary, NC, USA).

## 5. Conclusions

In conclusion, this phase 2a, placebo-controlled, randomized, clinical trial found that treatment with ALS-L1023 in patients with NAFLD reduced hepatic steatosis, stiffness, ALT, and total cholesterol levels without serious adverse events. This study suggests that ALS-L1023 can be a pharmacological treatment option for patients with NAFLD, without causing significant safety issues. Future studies investigating the efficacy and clinically appropriate dose of ALS-L1023 in a larger number of patients are called for.

## Figures and Tables

**Figure 1 pharmaceuticals-16-00623-f001:**
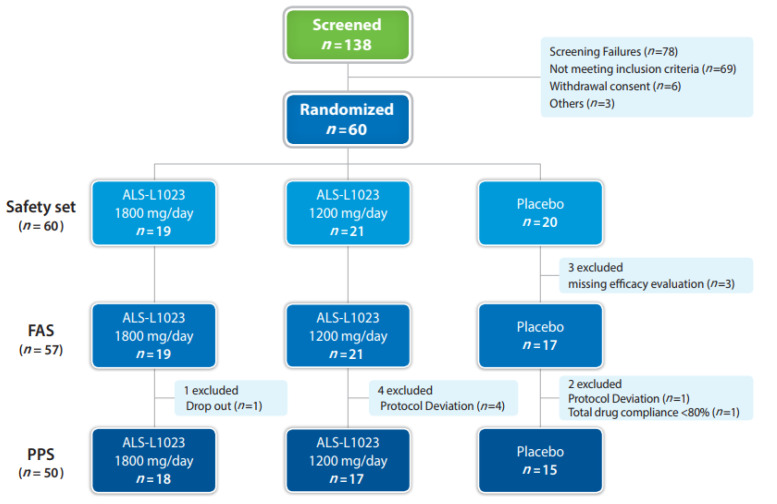
Clinical study flow diagram.

**Figure 2 pharmaceuticals-16-00623-f002:**
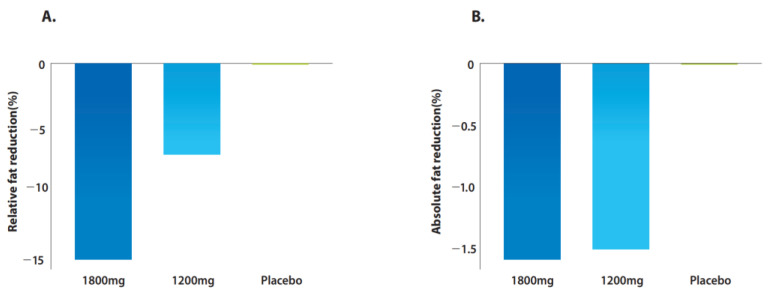
MRI-PDFF change in the FAS. MRI-PDFF results relative mean fat reduction percent from baseline (**A**), and absolute amount of fat reduction (**B**) as determined by MRI-PDFF at week 24.

**Figure 3 pharmaceuticals-16-00623-f003:**
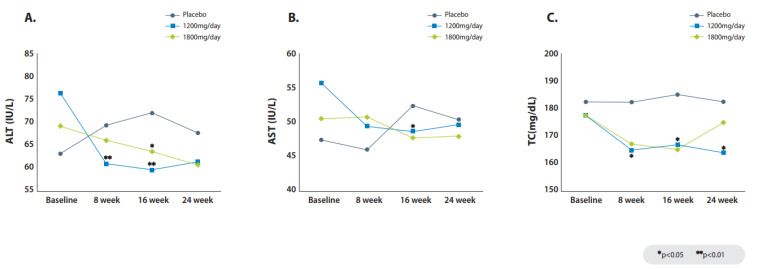
Change of liver biochemistry in the FAS. Time course of liver enzymes ALT (**A**), and AST (**B**). Time course of total cholesterol level (**C**).

**Figure 4 pharmaceuticals-16-00623-f004:**
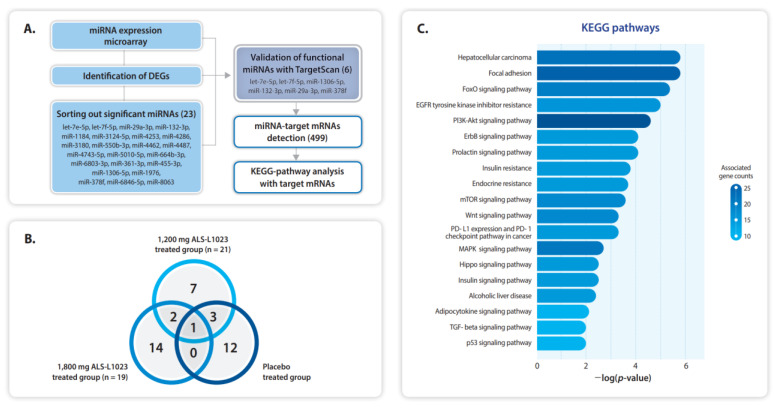
Putative miRNAs and target transcripts. (**A**) The schematic flow to select the putative miRNAs and their target transcripts. The numbers in parenthesis are significant miRNAs (23), putative miRNAs (6), and putative targets (499). (**B**) Venn diagrams showing the number of significant miRNAs from each administration group. (**C**) KEGG pathways of the putative targets.

**Table 1 pharmaceuticals-16-00623-t001:** Baseline Demographic and Clinical Characteristics.

Characteristic	ALS-L1023 1800 mg (*n* = 19)	ALS-L1023 1200 mg (*n* = 21)	Placebo (*n* = 20)	*p* ^1^
Demographic factors				
Age, mean ± SD, years	57.1 ± 13.8	51.2 ± 15.0	48.3 ± 14.1	0.15
Male sex, *n* (%)	12 (63.2)	11 (52.4)	12 (60.0)	0.78
Diabetes mellitus, *n* (%)	9 (47.4)	8 (38.1)	7 (35.0)	0.72
Hypertension, *n* (%)	12 (63.2)	6 (28.6)	8 (40.0)	0.08
Body mass index, median (Q1, Q3), kg/m^2^	29.7 (26.5, 31.6)	27.8 (26.0, 34.1)	27.5 (26.2, 31.0)	0.71
Serum biochemical levels				
ALT, median (Q1, Q3), U/L	54.0 (45.0, 89.0)	65.0 (42.5, 91.0)	52.5 (34.3, 80.5)	0.62
AST, median (Q1, Q3), U/L	46.0 (33.0, 60.0)	54.0 (33.5, 76.0)	44.5 (27.3, 64.5)	0.53
GGT, median (Q1, Q3), U/L	42.0 (34.0, 92.0)	70 (41.5, 109.5)	44.0 (26.5, 61.8)	0.05
ALP, median (Q1, Q3), U/L	80.0 (73.0, 95.0)	77.0 (67.0, 92.5)	83.5 (62.5, 111.0)	0.91
Total bilirubin, median (Q1, Q3), U/L	0.8 (0.6, 1.1)	0.8 (0.6, 0.8)	0.8 (0.6, 1.2)	0.71
Total cholesterol, median (Q1, Q3), mg/dL	168 (144, 208)	172 (156, 207)	182 (150, 206)	0.99
HDL cholesterol, median (Q1, Q3), mg/dL	48 (39, 58)	46 (38, 57)	46 (41, 52)	0.77
LDL cholesterol, median (Q1, Q3), mg/dL	107 (87, 123)	101 (83, 126)	113 (87, 136)	0.69
Triglycerides, median (Q1, Q3), mg/dL	169 (132, 225)	140 (111, 178)	188 (137, 232)	0.20
Fasting glucose, median (Q1, Q3)	109 (101, 135)	97 (89, 122)	98 (93, 119)	0.06
Fasting insulin, median (Q1, Q3)	17.1 (11.6, 28.0)	23 (12.2, 36.8)	15.6 (12.3, 24.9)	0.41
HOMA-IR, median (Q1, Q3)	4.14 (2.9, 7.0)	5.38 (3.0, 9.2)	4.18 (2.8, 5.9)	0.42
Imaging				
MRI-PDFF, median (Q1, Q3), %	14.6 (9.8, 21.5)	15.7 (10.6, 22.2)	15.7 (10.9, 20.3)	0.94
MRI-PDFF, mean ± SD, %	16.15 ± 6.1	17.4 ± 7.6	16.9 ± 7.4	
MRE, median (Q1, Q3), kPa	3.3 (2.7, 3.8)	3.0 (2.6, 4.5)	2.9 (2.6, 3.3)	0.51
MRE, mean ± SD, kPa	3.3 ± 0.7	3.7 ± 1.6	3.2 ± 0.8	
Common concomitant medications				
NSAIDs, *n* (%)	3 (15.8%)	1 (4.8%)	5 (25.0%)	0.18
Diabetes medicine, *n* (%)	9 (47.4%)	7 (33.3%)	7 (35.0%)	0.62
Proton pump inhibitors, *n* (%)	3 (15.8%)	5 (23.8%)	4 (20.0%)	0.92
Statins, *n* (%)	11 (57.9%)	14 (66.7%)	6 (30.0%)	0.05
Angiotensin-converting enzyme inhibitors, *n* (%)	6 (31.6%)	3 (14.3%)	6 (30.0%)	0.37

Abbreviations: ALP, alkaline phosphatase; ALT, alanine aminotransferase; AST, aspartate aminotransferase; GGT, gamma-glutamyl transferase; HDL, high-density lipoprotein; IQR, interquartile range; LDL, low-density lipoprotein; MRI-PDFF, MRI proton density fat fraction; MRE, MR elastography; Q1, quartile 1; Q3, quartile 3; SD, standard deviation. ^1^ Chi-squared test or Fisher’s exact test was used for categorical variables and ANOVA or Kruskal–Wallis test was used for continuous variables.

**Table 2 pharmaceuticals-16-00623-t002:** Change in MRI-proton density fat fraction from baseline to week 24.

	ALS-L1023 1800 mg		ALS-L1023 1200 mg		Placebo		*p* ^2^	*p* ^3^
FAS	Value	*p* ^1^	Value	*p* ^1^	Value	*p* ^1^		
PDFF, (relative) % change from baseline	−15.0 (22.0)	0.03	−6.9 (47.0)	0.49	0 (30.0)	0.23	0.95	0.65
PDFF, (absolute) change from baseline	−1.6 (4.6)	0.02	−1.5 (7.0)	0.41	0 (3.8)	0.34	0.48	0.85
Patients with ≥20% PDFF reduction	7 (38.9)		8 (40.0)		5 (29.4)		0.55	0.50
Patients with ≥15% PDFF reduction	9 (50.0)		9 (45.0)		5 (29.4)		0.21	0.33
Patients with ≥7% PDFF reduction	12 (66.7)		10 (50.0)		8 (47.1)		0.24	0.86
PPS	Value	*p* ^1^	Value	*p* ^1^	Value	*p* ^1^		
PDFF, (relative) % change from baseline	−15.0 (22.0)	0.03	−9.5 (42.6)	0.58	0.3 (43.1)	0.52	0.49	0.99
PDFF, (absolute) change from baseline	−1.6 (4.6)	0.02	−1.9 (6.8)	0.54	0.1 (5.1)	0.43		0.94
Patients with ≥20% PDFF reduction	7 (38.9)		7 (41.2)		4 (26.7)		0.46	0.39
Patients with ≥15% PDFF reduction	9 (50.0)		8 (47.1)		4 (26.7)		0.17	0.23
Patients with ≥7% PDFF reduction	12 (66.7)		9 (52.9)		6 (40.0)		0.13	0.46

Abbreviations: FAS, full analysis set; PPS, per protocol set; PDFF, proton density fat fraction. ^1^ Changes in the group analyzed by paired *t*-test or Wilcoxon signed-rank test. ^2^ Comparison between ALS-L1023 1800 mg and placebo groups analyzed by two-sample *t*-test, Wilcoxon rank-sum test, or Chi-squared test. ^3^ Comparison between ALS-L1023 1200 mg and placebo groups analyzed by two-sample *t*-test, Wilcoxon rank-sum test, or Chi-squared test.

**Table 3 pharmaceuticals-16-00623-t003:** Change in MRE from baseline to week 24.

	ALS-L1023 1800 mg		ALS-L1023 1200 mg		Placebo		*p* ^2^	*p* ^3^
FAS	Value	*p* ^1^	Value	*p* ^1^	Value	*p* ^1^		
MRE, (relative) % change from baseline	−6.2 (20.1)	0.23	−10.7 (17.3)	0.03	−7.6 (20.7)	0.14	0.66	0.20
MRE, (absolute) change from baseline	−0.3 (0.7)	0.10	−0.4 (0.7)	0.02	−0.2 (0.6)	0.12	>0.99	0.36
Patients with ≥20% MRE reduction	2 (11.1)		7 (35.0)		4 (23.5)		0.40	0.45
Patients with ≥15% MRE reduction	5 (27.8)		8 (40.0)		5 (29.4)		>0.99	0.50
Patients with ≥7% MRE reduction	8 (44.4)		14 (70.0)		9 (52.9)		0.62	0.29
PPS	Value	*p* ^1^	Value	*p* ^1^	Value	*p* ^1^		
MRE, (relative) % change from baseline	−6.2 (20.1)		−10.1 (17.2)		−11.9 (31.4)	0.16	0.62	0.53
MRE, (absolute) change from baseline	−0.3 (0.7)		−0.4 (0.6)		−0.4 (0.8)	0.17	0.86	0.51
Patients with ≥20% MRE reduction	2 (11.1)		7 (41.2)		4 (26.7)		0.37	0.39
Patients with ≥15% MRE reduction	5 (27.8)		7 (41.2)		5 (33.3)		1.00	0.65
Patients with ≥7% MRE reduction	8 (44.4)		12 (70.6)		9 (60.0)		0.37	0.53

Abbreviations: FAS, full analysis set; MRE, MR elastography; PPS, per protocol set. ^1^ Changes in the group analyzed by paired *t*-test or Wilcoxon signed-rank test. ^2^ Comparison between ALS-L1023 1800 mg and placebo groups analyzed by two-sample *t*-test, Wilcoxon rank-sum test, or Chi-squared test. ^3^ Comparison between ALS-L1023 1200 mg and placebo groups analyzed by two-sample *t*-test, Wilcoxon rank-sum test, or Chi-squared test.

**Table 4 pharmaceuticals-16-00623-t004:** Change in serum biomarkers from baseline to week 24 in the FAS.

	ALS-L1023 1800 mg		ALS-L1023 1200 mg		Placebo		*p* ^2^	*p* ^3^
Value	% CFB	*p* ^1^	% CFB	*p* ^1^	% CFB	*p* ^1^		
ALT, U/L	−12.4 (−30.2, 14.3)	0.09	−29.8 (−47.2, 21.9)	0.12	−4.9 (−15.4, 27.6)	0.50	0.11	0.10
AST, U/L	0 (−21.7, 25.0)	0.56	−19.6 (−41.3, 15.6)	0.13	0 (−16.1, 34.6)	0.61	0.44	0.12
GGT, U/L	−7 (−17.0, 15.8)	0.37	−6.3 (−40.6, 15.3)	0.30	1.0 (−9.0, 22.6)	0.92	0.43	0.35
ALP, U/L	1.2 (−7.4, 12.5)	0.90	−1.7 (−12.1, 5.9)	0.27	−3.6 (−19.2, 12.7)	0.08	0.18	0.31
Total bilirubin, mg/dL	−5.9 (−16.9, 25.0)	0.79	13.9 (−13.6, 33.0)	0.07	17.2 (−12.3, 60.2)	0.28	0.44	0.91
Triglycerides, mg/dL	−15.5 (−29.6,29.9)	0.16	−2.8 (−27.6, 11.4)	0.49	−21.3 (−32.1, −5.7)	0.001	0.55	0.05
TC, mg/dL	0.9 (−9.1,7.6)	0.60	−5.5 (−12.9, 0.0)	<0.01	5.4 (−6.3, 7.5)	0.59	0.52	0.03
HDL-C, mg/dL	2.2 (−9.1, 7.7)	0.97	−3.3 (−14.0, 11.5)	0.57	2.0 (−4.0, 15.5)	0.36	0.48	0.31
LDL-C, mg/dL	−4.1 (−10.1, 9.3)	0.32	−4.9 (−21.5, 4.5)	0.04	5.1 (−5.4, 11.1)	0.82	0.45	0.15
Pro-C3	1.0 (−6.6, 19.0)	0.62	−2.0 (−12.6, 23.5)	0.44	0 (−5.3, 17.6)	0.63	0.99	0.64
CK-18	14.2 (−34.0, 47.0)	0.59	0.4 (−45.8, 32.3)	0.47	−12.9 (−23.2, 87.3)	0.75	0.61	0.79
HOMA-IR	−2.3 (−25.7, 39.9)	0.72	−1.0 (−37.9, 45.2)	0.54	5.0 (−28.0, 33.8)	0.86	0.85	0.75
Leptin	−5.9 (−23.8, 15.9)	0.62	2.5 (−14.6, 28.6)	0.99	−5.0 (−16.8, 10.2)	0.43	0.55	0.75
Ghrelin	26.8 (0, 152.1)	0.01	34.8 (−12.7, 76.4)	0.05	17.2 (0.0, 90.5)	0.03	0.74	0.49
Adiponectin	7.4 (−21.6, 13.6)	0.44	−5.7 (−17.6, 20.8)	0.63	8.6 (−8.5, 5.9)	0.61	0.90	0.70
NFS	11.4 (−30.2, 29.1)	0.30	−13.6 (−27.0, 14.4)	0.08	0.9 (−22.5, 28.3)	0.63	0.27	0.64
Visceral Fat mass	−8.9 (−30.2, 9.8)	0.16	−4.4 (−15.4, 15.1)	0.94	−4.9 (−24.5, 10.9)	0.34	0.49	0.41

Abbreviations: ALP, alkaline phosphatase; ALT, alanine aminotransferase; AST, aspartate aminotransferase; CFB, change from baseline; CK-18, cytokeratin-18; GGT, gamma-glutamyl transferase; HDL, high-density lipoprotein; LDL, low-density lipoprotein; NFS, NAFLD fibrosis score; PRO-C3, N-terminal propeptide of type 3 procollagen; TC, total cholesterol. Values are presented in median (interquartile range). ^1^ CFB in the group was analyzed by paired *t*-test or Wilcoxon signed-rank test. ^2^ Comparison of CFBs in ALS-L1023 1800 mg and placebo groups by two-sample *t*-test or Wilcoxon rank-sum test. ^3^ Comparison of CFBs in ALS-L1023 1200 mg and placebo groups by two-sample *t*-test or Wilcoxon rank-sum test.

**Table 5 pharmaceuticals-16-00623-t005:** Safety.

	ALS-L1023 1800 mg (*n* = 19)	ALS-L1023 1200 mg (*n* = 21)	Placebo (*n* = 20)
Safety overview			
Any adverse events	8 (42.1)	13 (61.9)	8 (40.0)
Grade 3–4 adverse events	0 (0)	0 (0)	0 (0)
Serious adverse events	0 (0)	0(0)	1 (5.0)
Adverse event leading to discontinuation of any study drug	0 (0)	0(0)	1 (5.0)
Death	0 (0)	0 (0)	0 (0)
Most common adverse events			
Fatigue	1(5.3)	0 (0)	1 (5.0)
Dyspepsia	1(5.3)	3 (14.3)	2 (10.0)
Abdominal distension	0 (0)	2 (9.5)	0 (0)
Abdominal pain	1 (5.3)	1 (4.8)	0 (0)
Dry eye	1 (5.3)	0 (0)	0 (0)
Pulpitis dental	0 (0)	1 (4.8)	0 (0)
Nasopharyngitis	1 (5.3)	0 (0)	1 (5.0)
Urinary tract infection	1 (5.3)	3 (14.3)	2 (10.0)
Chest discomfort	1 (5.3)	1 (4.8)	1 (5.0)
Intervertebral disc disorder	1 (5.3)	0 (0)	0 (0)

## Data Availability

Data sharing is not applicable to this article.

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
