# Peer review of "A Phase 2a, Randomized, Double-Blind, Placebo-Controlled Study to Assess the Efficacy and Safety of ALS-L1023 in Non-Alcoholic Fatty Liver Disease"

_pharmaceuticals, 2023, doi:10.3390/ph16040623_

Round 1
Reviewer 1 Report
The study is original, well structured, rich in results and the conclusions are suitable.
However, the study has limitations that were well explained in the discussion.
I recommend the following changes:
-Please, in numbers in the order of thousands, do not insert the comma (1,200 mg, 1,800 mg): uniform throughout the manuscript
-Please, serum must be used in the singular: insert "in patient serum" (line 177)
-Fig.4: Please, arrange the images differently so that the writing can also be read in the printed manuscript
-Line 202: delete “the following”
-Line 208: alanine aminotransferase only needs to be entered once in full; use ALT
Author Response
Reviewer 1
The study is original, well structured, rich in results and the conclusions are suitable.
However, the study has limitations that were well explained in the discussion.
I recommend the following changes:
1. Please, in numbers in the order of thousands, do not insert the comma (1,200 mg, 1,800 mg): uniform throughout the manuscript
Response:
We appreciate the comment. We’ve revised it throughout the manuscript accordingly.
2.Please, serum must be used in the singular: insert "in patient serum"(line 177)
Response:
The manuscript was revised accordingly.
3. Fig.4: Please, arrange the images differently so that the writing can also be read in the printed manuscript
Response:
We’ve added a comment on Figure 4B so that the manuscript can be read as the images in Figure 4 is read:
Line 182: The numbers of significant miRNAs from each study group are presented in Figure 4B and the number of putative miRNAs was 6.
4. Line 202: delete “the following”
Response:
We have revised the manuscript as commented.
5. Line 208: alanine aminotransferase only needs to be entered once in full; use ALT
Response:
We have revised the manuscript as commented.

Reviewer 2 Report
In this study, Kim et al. tested the efficacy of ALS-L1023, a herbal extract from Melissa officinalis in the treatment of non-alcoholic fatty liver disease (NAFLD). This phase 2a clinical trial involved 19 patients receiving 1,800 mg ALS-L1023, 21 patients receiving 1,200 mg ALS-L1023, and 17 patients receiving a placebo. The study took 24 weeks. Efficacy was assessed through the measurement of factors including hepatic fat, liver stiffness, and hepatic enzymes. The ALS-L1023 group has shown reductions in hepatic fat levels, live stiffness, and serum alanine aminotransferase. ALS-L1023 reduced hepatic fat content without any obvious adverse effects. Moreover, downregulation of 23 microRNA was detected in ALS-L1023-treated patients. These microRNAs have been related to signaling pathways that are upregulated in NAFLD. Their downregulation by ALS-L1023 suggests that this herbal extract interferes with cell signaling alterations related to the disease pathology. ALS-L1023 has also been characterized as an anti-angiogenic agent, putting forward angiogenesis inhibition as a potential strategy for the treatment of NAFLD.
Specific recommendations to the authors:
1. In Table 1, it may not the obvious to the reader that "p" is the p-value. It is suggested, in accordance with the other Tables, to mark the "p" in Table 1 with a superscript number to link it to the statistical tests used herein.
2. In Tables S1-S3, please correct "Change" to "Changes".
3. In Table S4, please correct "Significant changed" to "Significantly changed".
4. Please include the full names of "Pro-C3" and "CK-18" in Table 4 and in the main text.
5. In Figures 2A-B and 3A-C, please make the axes visible in the graphs. Please consider eliminating the background and the grid lines of the graphs.
6. Please mention Figure 4B in the main text.
7. In the legend of Figure 4C, the sentence "Venn diagrams showing the number of 196 significant miRNAs from." sounds incomplete. Please complete it.
Author Response
1. In Table 1, it may not the obvious to the reader that "p" is the p-value. It is suggested, in accordance with the other Tables, to mark the "p" in Table 1 with a superscript number to link it to the statistical tests used herein.
Response: We appreciate the thorough review of our manuscript. We have revised the manuscript as commented.
2. In Tables S1-S3, please correct "Change" to "Changes".
Response: As the reviewer’s comment, we have revised it accordingly.
3. In Table S4, please correct "Significant changed" to "Significantly changed".
Response: We have revised the manuscript as commented.
4. Please include the full names of "Pro-C3" and "CK-18" in Table 4 and in the main text.
Response: We have revised the manuscript as commented.
5. In Figures 2A-B and 3A-C, please make the axes visible in the graphs. Please consider eliminating the background and the grid lines of the graphs.
Response: Figures 2 and 3 along with supplementary figures 2 and 3 were revised as commented.
6. Please mention Figure 4B in the main text.
Response: We’ve added a comment on Figure 4B in the revised manuscript as follows:
Line 182: The numbers of significant miRNAs from each study group are presented in Figure 4B and the number of putative miRNAs was 6.
7. In the legend of Figure 4C, the sentence "Venn diagrams showing the number of 196 significant miRNAs from." sounds incomplete. Please complete it.
Response: We have revised it.
Line 195: Figure 4. Putative miRNAs and target transcripts. (A) The schematic flow to select the putative miRNAs and their target transcripts. The numbers in parenthesis are significant miRNAs (23), putative miRNAs (6), and putative targets (499). (B) Venn diagrams showing the number of significant miRNAs from each administration group. (C) KEGG pathways of the putative targets.

Reviewer 3 Report
1. Please double check on reference 12 and 19. They did not match with the description in the manuscript.
2. Please clarify how the the treatment dosage of ALS-L1023 was determined and include relevant references.
3. Did the authors examine if gender and hormonal differences may affect the result of the study? Most of the previous studies used male animals with only one study using female ovariectomized mice.
4. Based on the study results, ALS-L1023 at 1200 mg or 1800 mg have different effects on hepatic steatosis, inflammation and fibrosis. What is the authors' interpretation?
5. Line 190-192. Content could be moved to the Discussion.
6. For serum biomarker analysis, how were the blood samples collected from all participants?
7. Please read over carefully to check grammatical errors and consistency.
Author Response
1. Please double check on reference 12 and 19. They did not match with the description in the manuscript.
Response: Thank you for your thorough review of our manuscript. References 12 and 19 were deleted and reference 11 replaced reference 12.
Line 48: Studies have found that it has antioxidant activity and protects cells against oxidative stress-induced apoptosis [11].
2. Please clarify how the treatment dosage of ALS-L1023 was determined and include relevant references.
Response: On rats, no toxicity was observed in a test in which 2,000 mg/kg was administered and no subject died in a test with 13-week administration of 500, 1000, and 2,000 mg/kg per day. There were no clinical symptoms or systemic toxicity related to the administration. On beagle dogs, no toxicity was observed in which 250, 500, and 1,000 mg/kg per day was administered for 2 weeks. No toxicity was found even with the highest concentration of 1,000 mg/kg per day.
When the harmless dose of 2,000 mg/kg per day in rats and that of 1,000 mg/kg per day in beagle dogs are converted into Human Equivalent Dose (HED), they are 19,354 mg and 33,333 mg per day for a 60 kg adult. Therefore, the safety dosage was calculated as 19,354 mg or less in adult patients.
The dosage effectiveness was determined by referencing the dosage that showed the effect of ALS-L1023 in an animal model of fatty liver induced by a high-fat diet. The effective dosages in the HFD mouse model were 100 and 200 mg/kg per day which can be converted to 487 and 975 mg per day for a 60 kg adult.
In a recent study, two doses of low (800 mg/kg per day) and high (1,200 mg/kg per day) doses of ALS-L1023 were selected and mixed with feed for administration to C57BL/6 wild-type male mice with NAFLD. The study found that ALS-L1023 may exert anti-fibrotic effects in NAFLD, suggesting potential benefits as a treatment of liver fibrosis. [EJ Lee et al. Life (Basel). 2022 Dec 29;13(1):100. doi:10.3390/life13010100.]
In a phase 2 clinical trial for patients with abdominal obesity, which has not been published yet though, efficacy and safety have been confirmed with 600 mg and 1,200 mg per day which were +20% of the converted HED after considering the ±20% interval of the standard conversion factor.
Therefore, in this clinical trial, we aimed to assess the performance of ALS-1023 with the dosage of 1,200 mg since it is a previously verified effective dose and with that of 1,800 mg since it is the secured safety margin.
3. Did the authors examine if gender and hormonal differences may affect the result of the study? Most of the previous studies used male animals with only one study using female ovariectomized mice.
Response: This study could not perform subgroup analysis by gender since the number of patients was rather small. Future studies involving more patients may carry out subgroup analysis by gender and find whether hormonal differences of the study patients affect the result. The revised manuscript has it mentioned as a limitation.
Line 257: Subgroup analysis by gender could not be performed since the number of patients was rather small.
4. Based on the study results, ALS-L1023 at 1200 mg or 1800 mg have different effects on hepatic steatosis, inflammation and fibrosis. What is the authors' interpretation?
Response: The authors take it as a need for future studies. Although different dosages affected hepatic steatosis, inflammation, and fibrosis differently, we are in no position of providing an interpretation yet.
5. Line 190-192. Content could be moved to the Discussion.
Response: We have revised it accordingly.
Line 242: ALS-L1023 treatment could have regulated the abundance of miRNAs in the serum, thereby resulting in changes in cellular signaling.
6. For serum biomarker analysis, how were the blood samples collected from all participants?
Response: The blood samples of all the study patients were collected at baseline at each participating center.
7. Please read over carefully to check grammatical errors and consistency.
Response: We had the entire manuscript proofread by a professional English proofreading agency. And all the changes were underlined and marked.
